# Seed Priming with MeJa Prevents Salt-Induced Growth Inhibition and Oxidative Damage in *Sorghum bicolor* by Inducing the Expression of Jasmonic Acid Biosynthesis Genes

**DOI:** 10.3390/ijms241210368

**Published:** 2023-06-20

**Authors:** Takalani Mulaudzi, Gershwin Sias, Mulisa Nkuna, Nzumbululo Ndou, Kaylin Hendricks, Vivian Ikebudu, Abraham J. Koo, Rachel F. Ajayi, Emmanuel Iwuoha

**Affiliations:** 1Life Sciences Building, Department of Biotechnology, University of the Western Cape, Private Bag X17, Bellville 7535, South Africa; 3539430@myuwc.ac.za (G.S.); 4078438@myuwc.ac.za (M.N.); 3992677@myuwc.ac.za (N.N.); 3338080@myuwc.ac.za (K.H.); 3206244@myuwc.ac.za (V.I.); 2SensorLab, Department of Chemical Sciences, University of the Western Cape, Private Bag X17, Bellville 7535, South Africa; rngece@uwc.ac.za (R.F.A.); eiwuoha@uwc.ac.za (E.I.); 3Department of Biochemistry, University of Missouri, Columbia, MO 65211, USA; kooaj@missouri.edu

**Keywords:** gene expression, FTIR, MeJa, photosynthetic pigments, priming, salt stress, *Sorghum bicolor*, osmotic adjustment, oxidative damage

## Abstract

Salinity is one of the major detrimental abiotic stresses at the forefront of deterring crop productivity globally. Although the exogenous application of phytohormones has formerly proven efficacious to plants, their effect on the moderately stress-tolerant crop “*Sorghum bicolor*” remains elusive. To investigate this, *S. bicolor* seeds primed with methyl jasmonate (0; 10 and 15 μM MeJa) were exposed to salt (200 mM NaCl) stress, and their morpho-physiological, biochemical, and molecular attributes were measured. Salt stress significantly decreased shoot length and fresh weight by 50%, whereas dry weight and chlorophyll content were decreased by more than 40%. Furthermore, salt-stress-induced oxidative damage was evident by the formation of brown formazan spots (indicative of H_2_O_2_ production) on sorghum leaves and a more than 30% increase in MDA content. However, priming with MeJa improved growth, increased chlorophyll content, and prevented oxidative damage under salt stress. While 15 µM MeJa maintained proline content to the same level as the salt-stressed samples, total soluble sugars were maintained under 10 µM MeJa, indicating a high degree of osmotic adjustment. Shriveling and thinning of the epidermis and xylem tissues due to salt stress was prevented by MeJa, followed by a more than 70% decrease in the Na^+^/K^+^ ratio. MeJa also reversed the FTIR spectral shifts observed for salt-stressed plants. Furthermore, salt stress induced the expression of the jasmonic acid biosynthesis genes; *linoleate 92-lipoxygenase 3, allene oxide synthase 1, allene oxide cyclase,* and *12-oxophytodienoate reductase 1*. In MeJa-primed plants, their expression was reduced, except for the *12-oxophytodienoate reductase 1* transcript, which further increased by 67%. These findings suggest that MeJa conferred salt-stress tolerance to *S. bicolor* through osmoregulation and synthesis of JA-related metabolites.

## 1. Introduction

Sorghum is a sought-after cereal crop in Africa due to its relatively high drought-tolerant characteristics, digestibility, and rich nutritional value [1,2]. The high levels of tannins and phenolic compounds found especially in the red and brown grain varieties are highly beneficial and protect humans against diabetes, cardiovascular, and hypertensive diseases [3]. As a staple food globally, sorghum also serves as a feedstock, fiber, and a source of biofuel [4]. Therefore, its production will not only improve food security but will also be highly beneficial for human health and in solving the rising energy crisis [2]. The sorghum genome comprises a relatively minuscule 730 million base pairs, which makes it a vital model organism for the comparative genomic studies of other C4 crops [5]. Additionally, sorghum’s high inbreeding level and gene flow are highly attractive characteristics in studying genetic systems. Plants play a considerable role in the agricultural sector that benefits the livelihoods of many individuals worldwide and is continuously expanding. The agriculture, forestry, and fisheries sectors contributed to the 1.3% growth for South Africa’s economy experienced in 2017, exceeding the expected 1% [6,7]. Therefore, agricultural sustainability should be prioritized to capitalize on its potential in boosting the world economy. 

Regardless of the economic gain, the growing population and challenges associated with climate change are continuously putting severe strain on food security. Thus, food production is required to increase by an estimated 70–100% in order to feed the expected ~9.8 billion global population by the year 2050 [8]. Harsh environmental conditions are the leading cause of low crop productivity and have resulted in a more than 50% reduction in crop yield globally [9]. Salinity has proven to be one of the most severe and tenacious abiotic stressors due to the increased levels of salts found in arable lands globally. Many plant species succumb when exposed to NaCl concentrations exceeding 200 mM, due to induced oxidative stress, which is caused by the overproduction of reactive oxygen species (ROS) [10]. Reactive oxygen species, including singlet oxygen (^1^O_2_), superoxide radical (O_2_**^∙−^**), hydrogen peroxide (H_2_O_2_), and hydroxyl radical (^∙^HO), cause damage to lipids, proteins, carbohydrates, and nucleic acids when produced in high concentrations [11,12]. Malondialdehyde (MDA) is one of the major products of lipid peroxidation and has therefore been extensively accepted as an oxidative stress marker and for assaying the level of lipid peroxidation [13]. Furthermore, salinity causes ion toxicity in cells due to the absorption and accumulation of toxic (Na^+^ and Cl^−^) ions, while negatively preventing the absorption of essential elements such as K^+^, P^+^, Mg^+^, and Si^+^ [14]. Ion toxicity leads to nutrient imbalances, loss of membrane function, and decreased photosynthesis, and also affects stomatal conductance and the plant’s ability to scavenge ROS [15]. However, plants, use either exclusion or compartmentalization of these toxic ions as part of their survival strategy, and this is mostly represented as the Na^+^/K^+^ ratio [16,17]. The damaging effects of salt stress on plant growth and development have been reported in several plant species such as *Solanum lycopersicum* L. [18], *Zingiber officinale* [19], and *Gossypium hirsutum* L. [20]. The salt-induced changes in plant metabolism regulate the plant’s ROS scavenging capability by altering antioxidant enzyme activities [16,21] and triggering a cascade of signaling networks inducing the expression of stress-responsive genes [22,23]. Plants also counteract stressors by synthesizing metabolites such as polyamines, proline, and soluble sugars to stabilize the osmotic potential, inhibit the over-accumulation of free radicals, and conserve cellular metabolism by protecting cell membranes and proteins [24]. The increase in cellular osmolality due to the induced production of osmolytes is another strategy employed by cells to maintain adequate water to allow the necessary turgor for cell expansion [25].

Phytohormones are essential regulators of plant response and may act synergistically. They also have antagonistic effects on plant growth and development, as observed with the role of ethylene in inhibiting vegetative growth through its interaction with auxins [24,26]. At physiological levels, plant hormones are produced in low concentrations and they regulate diverse plant cellular processes. However, stressful conditions induce their production, which signal downstream developmental processes and therefore alleviate injury, thus providing tolerance to plants [27,28]. Plant growth and yield depend on auxins, cytokinins, and gibberellins, whereas enhanced stress response largely involves the systemic interactions between abscisic acid, salicylates, and jasmonates [29]. Jasmonic acid (JA) plays a key role in a number of developmental processes, including seed germination, plant maturity, and senescence. High concentrations of JA and its derivatives (jasmonate isoleucine conjugate (JA-Ile) and methyl jasmonate (MeJA)) have been reported in halophytes and glycophytes that were exposed to salt stress [30], suggesting their involvement in salt stress tolerance.

Jasmonic acid biosynthesis is mediated by the action of enzymes including lipoxygenase (LOX), allene oxide synthase (AOS), allene oxide cyclase (AOC), and oxophytodienoic acid reductase (OPR). Their expression is often induced by stress stimuli due to their importance in regulating stress tolerance in plants [30,31]. Extensive research exists on the effect of jasmonates in plants under biotic and abiotic stresses. However, the role and mechanism of MeJa-induced salinity tolerance in moderately stress-tolerant crops such as *S. bicolor* remains elusive. Methyl Jasmonate is an excellent activator of plant response to stress stimuli due to its hydrophobicity and volatile nature, making it easier to pervade the cytoplasm and trigger plant response [31]. Several studies successfully demonstrated the expression of genes involved in important pathways in cereal crops including Salt Overlay Sensitive [16,17], sodium-proton antiporter, antioxidant [22,32], and the JA biosynthesis pathways [33,34,35]. This study aims to investigate for the first time the role of exogenous MeJa in alleviating the negative effects of salt stress in *S. bicolor* plants. This was achieved by priming *S. bicolor* seeds with low concentrations of MeJa before exposing them to salt stress, and the responses were evaluated by assaying morpho-physiological traits such as shoot length, biomass, and photosynthetic pigments, whereas biochemical traits included oxidative stress markers (H_2_O_2_ and MDA), osmolytes (proline and total soluble sugars), anatomical structure, element content, and nature of biomolecules. The molecular effects of MeJa in alleviating salt stress effects in *S. bicolor* plants were elucidated by assaying the expression of JA biosynthesis genes.

## 2. Results

### 2.1. Morpho-Physiological Response of S. bicolor to MeJa under Salt Stress

To understand the role of priming with MeJa in alleviating the effects of salt stress in *S. bicolor*, three morpho-physiological traits including, growth (shoot length), biomass (fresh and dry weight), and photosynthetic pigment were assayed from plants primed with 10 µM and 15 µM MeJa before exposing to 200 mM NaCl.

#### 2.1.1. Growth and Biomass

The detrimental effect of salt stress on *S. bicolor*’s morphological attributes was evident as shown in Figure 1 and Table 1. Under salt stress, shoot length decreased by 53% from 401 mm to 188 mm (Figure 1A,B). Priming with MeJa led to improvements in the growth of *S. bicolor* under salt stress (200 mM NaCl), where priming with 10 μM and 15 μM MeJa resulted in a 68% and 77% increase in shoot length, respectively. Control plants (without NaCl treatment) primed with MeJa (10 μM and 15 μM) showed no significant differences compared to unprimed plants (Appendix A).

The effects of salt stress on the growth of *S. bicolor* plants were also assessed based on fresh weight (FW) and dry weight (DW) as shown in Table 1. Fresh weight decreased by 50% from 0.8 g to 0.397 g in *S. bicolor* plants stressed with 200 mM NaCl. However, priming with MeJa resulted in 84% (10 µM MeJa) and 68% (15 µM MeJa) increases in FW. Salt stress also decreased *S. bicolor*’s DW by 42% from 0.091 g to 0.053 g in *S. bicolor* plants stressed with 200 mM NaCl. Priming with MeJa improved DW, and this resulted in 109% (10 μM MeJa) and 94% (15 μM MeJa) increases for *S. bicolor* plants stressed with 200 mM NaCl as compared to unprimed plants (Table 1). Priming with MeJa (10 µM and 15 µM) showed no significant effects on the FW and DW of control *S. bicolor* plants (Appendix A).

#### 2.1.2. Photosynthetic Pigments

Salt stress slightly impacted chlorophyll content, but these effects were reversed by priming with MeJa (Table 2). Salt stress decreased chlorophyll *a* content by 47% from 8.17 mg/gFW to 4.308 mg/gFW. However, priming with 10 µM MeJa and 15 µM MeJa increased chlorophyll *a* content by 178% (11.992 mg/gFW) and 111% (9.104 mg/gFW) in *S. bicolor* plants stressed with 200 mM NaCl. No significant effect was observed in chlorophyll *b* content for all treatments. Salt stress significantly decreased the total chlorophyll content of *S. bicolor* plants by 44% from 11.96 mg/gFW to 6.667 mg/gFW. Priming with MeJa increased total chlorophyll content in salt-stressed plants by 151% (16.710 mg/gFW) and 102% (13.438 mg/gFW) when primed with 10 µM and 15 µM MeJa, respectively. Priming with MeJa (10 µM and 15 µM) showed no significant effects on the photosynthetic pigments in control (without NaCl treatment) plants (Appendix A).

### 2.2. Biochemical Response of S. bicolor to MeJa under Salt Stress

The biochemical response of *S. bicolor* to MeJa (10 µM and 15 µM) priming under salt (200 mM NaCl) stress was determined by assaying the content of oxidative stress markers (H_2_O_2_ and MDA), osmolytes (proline and total soluble sugars), anatomic structure, element distribution and the structure of biomolecules, which were analyzed using FTIR spectra.

#### 2.2.1. ROS Production

Salt stress induced the accumulation of H_2_O_2_ as indicated by increased levels of brown formazan (indicative of H_2_O_2_ formation) precipitation on *S. bicolor* leaves in response to 200 mM NaCl (Figure 2A). Priming with 10 µM and 15 µM MeJa resulted in substantial reductions in the formation of brown formazan on the leaves of *S. bicolor* plants exposed to 200 mM NaCl (Figure 2A). Priming with 10 µM and 15 µM MeJa showed no significant effects on the formation of H_2_O_2_ in control (without NaCl treatment) plants (Appendix A).

Lipid peroxidation was assayed based on the MDA content, which increased by 46% from 17.94 nmol/g FW to 26.15 nmol/g FW in *S. bicolor* plants stressed with 200 mM NaCl (Figure 2B). In *S. bicolor* plants grown from seeds primed with MeJa, MDA content decreased from 26.15 nmol/g FW MDA to 20.34 nmol/g FW (10 µM MeJa) and 24.06 nmol/g FW (15 µM MeJa) (Figure 2B). Significant decreases in MDA content were observed in *S. bicolor* plants grown from seeds primed with 10 µM MeJa, indicating a 22% decrease. Furthermore, non-significant reductions in MDA content were observed for 10 µM and 15 µM MeJa priming for control (without NaCl treatment) plants (Appendix A).

#### 2.2.2. Proline and Total Soluble Sugars Content

Salt stress significantly increased proline content in *S. bicolor* plants compared to control plants (Figure 3A). Proline content increased by 203% from 2.27 µmol/gFW to 6.88 µmol/gFW when *S. bicolor* plants were stressed with 200 mM NaCl (Figure 3A). *S. bicolor* plants grown from seeds primed with MeJa resulted in a 27% decrease in proline content from 6.88 µmol/gFW to 4.84 µmol/gFW (10 µM MeJa) and 7.47 µmol/gFW (15 µM MeJa), indicating a 27% decrease for 10 µM MeJa-primed seeds. Priming with MeJa (10 µM and 15 µM) showed no significant effects on proline content in control (without NaCl treatment) plants (Appendix A).

Salt stress increased total soluble sugar (TSS) content in *S. bicolor* plants (Figure 3B), by 159% from 0.296 mg/g FW to 0.767 mg/g FW. Furthermore, MeJa pre-treated *S. bicolor* plants resulted in TSS content of 0.856 mg/g FW (10 µM MeJa) and 0.575 mg/g FW (15 µM MeJa) when exposed to 200 mM NaCl. However, only 15 µM MeJa was significant and led to a 25% decrease in TSS for salt-stressed *S. bicolor* plants. Priming with MeJa was not significant in control (without NaCl treatment) plants (Appendix A).

#### 2.2.3. Anatomic Structure and Element Analysis

Scanning electron microscope micrographs of *S. bicolor*’s epidermis (blue arrows) and xylem (red arrows) layers indicated slightly rough and shrunken layers under 200 mM NaCl treatment (Figure 4B) as compared to control (without NaCl treatment) plants (Figure 4A). Furthermore, the xylem (red arrows) walls under salt stress treatment showed thinning with an oval shape (Figure 4B, see the zoomed-in area), whereas the xylem walls from the control (without NaCl treatment) plants showed a round shape. This is even clearer when looking at the zoomed-in area (Figure 4A). However, *S. bicolor* plants grown from seeds primed with 10 µM and 15 µM MeJa before being exposed to salt stress showed a considerably smooth epidermis (blue arrows), while the thickness of the xylem walls was restored (Figure 4C,D) when compared to those plants under salt stress only (Figure 4B).

The element distribution, particularly Na^+^, K^+^ and Si^+^ in *S. bicolor* plants grown from seeds primed with MeJa and exposed to 200 mM NaCl, was analyzed using Scanning Electron Microscopy-Energy Dispersive X-ray Spectroscopy (SEM-EDX) (Figure 4E–H; Table 3). The Na^+^ content increased by 100% in *S. bicolor* plants exposed to 200 mM NaCl stress, while K^+^ and Si^+^ decreased by 80% and 100%, respectively. Salt negatively impacted the absorption and distribution of nutrients in *S. bicolor* plants, as seen by a rise in the Na^+^/K^+^ ratio of 1.4 (Figure 4F; Table 3). *S. bicolor* plants grown from seeds primed with MeJa before salt stress treatment promoted K^+^ absorption, as shown by the decrease in Na^+^/K^+^ ratio of 0.4 under 10 μM MeJa (Figure 4G) and 0.16 under 15 μM MeJa priming (Figure 4H), indicating 71% and 89% decreases in the Na^+^/K^+^ ratio, respectively. MeJa also positively influenced Si^+^ absorption, as shown by the 100% increase in Si^+^ under all MeJa concentrations (Table 3).

#### 2.2.4. Changes in Biomolecules

In this study, infrared radiation was used to identify changes caused in the active components of *S. bicolor* plants in response to pre-treatment with MeJa under 200 mM salt stress (Figure 5). Using FTIR technology, this study focused on a spectral region of 4000–500 cm^−1^. FTIR analysis identified 10 different functional groups across all samples as indicated by frequency ranges of 3700–3200 cm^−1^; 3100–2850 cm^−1^; 2140–2100 cm^−1^; 1740–1720 cm^−1^; 1680–1620 cm^−1^; 1400–1000 cm^−1^; 1320–1000 cm^−1^; 1250–1080; 910–665 cm^−1^; and 600–500 cm^−1^. Spectral peaks 3604, 3527, 3326, and 3295 represented the O-H stretching vibrations (3700–3200 cm^−1^), indicating the presence of different forms of phenolic groups (aromatic compounds). Spectral peaks 3068 and 2931 represented C-H stretching vibrations found within the frequency range 3100–2850 cm^−1^, indicating the presence of alkanes. The peak at 1677 cm^−1^ under the frequency range 1680–1620 cm^−1^ represented C=C stretching vibration. Indicating the presence of alkenes. Whereas the peak at 2135 cm^−1^ under the frequency range of 2140–2100 cm^−1^ represented the CC stretching vibrations, indicating the presence of alkynes which, together with alkenes and alkanes, represent aliphatic compounds such as carbohydrates. The peak at 1733 cm^−1^ represented C=O stretching vibration under the frequency range 1740–1720 cm^−1^, indicating the presence of aldehyde-containing aliphatic compounds. Spectral peaks at 1370 cm^−1^ represented C-F stretching vibration under the frequency range 1400–1000 cm^−1^, indicating the presence of alkyl-halide-containing fluoro compounds. The spectral peak at 1041 cm^−1^ represented C-O stretching vibration, which may be assigned to alcohols and lipids. The peak at 1250 cm^−1^ represented C-N stretching vibrations under frequency range 1250–1080 cm^−1^, indicating the presence of aliphatic-amine-containing proteins. Whereas peaks 824 cm^−1^ and 670 cm^−1^ represented N-H stretching vibrations under frequency range 910–665 cm^−1^, indicating the presence of proteins with primary and secondary amines. Lastly, the peak at 551 cm^−1^ represented C-F stretching vibration under frequency range 600–500 cm^−1^, indicating the presence of halo compounds. 

Salt stress disturbed the metabolism of *S. bicolor* by greatly altering the composition of organic molecules including phenolic compounds, carbohydrates, lipids, proteins, and secondary metabolites (Figure 5, red spectrum). Constant band patterns were observed throughout all samples analyzed. However, increased band absorption was observed in salt-stressed *S. bicolor* plants as compared to unstressed plants, which were evident at peaks 3295 (phenolics), 2135 (carbohydrates), 1250; 824 (proteins) and 551 cm^−1^ (halo compounds).

To understand the effect of MeJa in the composition and alteration of these organic biomolecules under salt stress, the infrared absorption spectra of *S. bicolor* plants pre-treated with 10 µM and 15 µM MeJa when subjected to 200 mM NaCl were analyzed. A major shift with increased absorption was observed in MeJa-pre-treated (10 µM MeJa) plants at the 3700–2750 cm^−1^ frequency range at maximum peak values of 3604 and 3295 cm^−1^, which represent phenolic compounds, whereas 3068 cm^−1^ represents carbohydrates. However, considerable decreases in absorption were observed at peaks 2135 cm^−1^ (lipids), 824 cm^−1^, 670 cm^−1,^ and 551 cm^−1^ (proteins) compared to salt-stressed plants. *S. bicolor* plants pre-treated with 15 µM MeJa resulted in slight decreases in the absorption intensity at 2135 cm^−1^ (lipids) and 824 cm^−1^ (proteins) compared to salt-stressed plants (Figure 5 green spectrum).

### 2.3. Gene Expression Analysis of Jasmonic Acid Biosynthesis Pathway Genes in S. bicolor under Salt Stress in Response to MeJa

The expression of genes encoding the JA biosynthesis pathway was analyzed using Real-time quantitative PCR (RT-qPCR) in *S. bicolor* plants grown from seeds primed with 10 µM and 15 µM MeJa under 200 mM NaCl stress. These genes included the (1) putative, *S. bicolor linoleate 9S-lipoxygenase 3* (*SbLOX*); (2) predicted, *S. bicolor allene oxide synthase 1*, *chloroplastic* (*SbAOS*); (3) predicted, *S. bicolor allene oxide cyclase*, *chloroplastic* (*SbAOC*); and (4) predicted, *S. bicolor 12-oxophytodienoate reductase 1* (*SbOPR*) (Table 4; Figure 6).

All the above-mentioned genes were constitutively expressed under physiological conditions in *S. bicolor* plants. Expression levels of *SbLOX* (Figure 6A) and *SbAOC* (Figure 6B) were relatively high, followed by *SbAOS* (Figure 6C) and lastly *SbOPR* (Figure 6D). The *SbLOX*, *SbAOS*, *SbAOC,* and *SbOPR* transcripts significantly increased by 95%, 65%, 145%, and 31%, respectively, in *S. bicolor* plants exposed to 200 mM NaCl stress. Under 200 mM NaCl stress, a significant downregulation of 42% and 58% in the transcript level of *SbLOX* was observed in *S. bicolor* plants grown from seeds primed with 10 µM and 15 µM MeJa, respectively. Pre-treatment with 15 µM MeJa resulted in a 44%, 63.9%, and 27% decrease in the transcript levels of *SbAOS*, *SbAOC,* and *SbOPR*, respectively, compared to the salt treatment only. However, *S. bicolor* plants grown from seeds primed with 10 µM MeJa resulted in a further 67% increase in the relative expression level of *SbOPR* under 200 mM NaCl stress (Figure 6D).

## 3. Discussion

Salt stress caused significant reductions in the growth and biomass of *S. bicolor* plants, observed as the decrease in shoot length, fresh weight (FW), and dry weight (DW) (Figure 1 and Table 1). Thus, the salt-induced growth reduction in *S. bicolor* plants might be due to the inhibition in cell division and elongation, decreased water uptake, energy diversion for survival, and a decrease in carbon [3,36]. The inhibitory effects of salt stress on the growth of plants are well documented and can also be observed with the decreasing growth of *S. bicolor* [36,37], *Solanum lycopersicum* L. [18], *Brassica napus* L. [38], *Chenopodium quinoa* [39], and *Triticum aestivum* L. [40] in response to salt stress. Pre-treatment with MeJa increased the plant biomass of *S. bicolor* plants under harsh environments (200 mM NaCl) as seen by the significant increase in shoot length, FW, and DW. The enhancing growth under salt stress may be due to MeJa’s role in inducing the plant’s survival strategies and JA’s role in promoting cell expansion [41]. This finding aligns with the results obtained in *Oryza sativa* L. under arsenic stress [42], *Cicer arietinum* L. [43], *Crithmum maritimum* L. [44], *Brassica napus* L. [38], and *Triticum aestivum* L. [45] under salt stress.

Chlorophyll is a vital pigment in photosynthesis and can indirectly be used as an indicator of the plant’s photosynthetic capacity and nutritional status [46]. These pigments are susceptible to nutrient deficiency and environmental stresses. To assess the effect of salt stress on the photosynthetic pigments, chlorophyll *a*, chlorophyll *b,* and total chlorophyll were measured in *S. bicolor* plants treated with 200 mM NaCl. Salt stress proved deleterious by significantly decreasing chlorophyll content in *S. bicolor* plants (Table 2) and therefore inhibiting plant photosystem [46,47]. The reduction in chlorophyll content could be indicative of restricted chlorophyll biosynthesis in conjunction with the induction of chlorophyllases [48]. Reduced chlorophyll content was also observed in *Calendula officinalis* L. and *Glycine max* [47,49] upon exposure to salt stress. The effect of MeJa on the chlorophyll content of *S. bicolor* was assessed in response to salt stress, and the results indicated that MeJa was able to reverse the deleterious effect of salt stress on chlorophyll content by significantly increasing chlorophyll content under salt stress (Table 2). The increased chlorophyll content in MeJa pre-treated *S. bicolor* plants in response to salt stress might be due to MeJa playing a role in the formation of 5-aminolevulinic acid, which is a vital enzyme in the biosynthesis of chlorophyll [50]. Similarly, improved chlorophyll content was observed in *Brassica napus* L. [38], *Anschusa italica* [51], *Citrus sinensis* [52], *Capsicum annuum* [53], and *Prunus dulcis* [54] when exposed to salt stress in response to MeJa treatment [55].

Maintaining the steady state of ROS molecules, including H_2_O_2_, is important for regulating many molecular mechanisms in plant cells, but over-accumulation is detrimental to plants causing severe oxidative damage [11]. Salt stress strongly affected ROS homeostasis by promoting the overproduction of H_2_O_2_ and therefore inducing oxidative damage in *S. bicolor* plants (Figure 2). This was illustrated by the over-accumulation of brown formazan, from a reaction catalyzed by horseradish peroxidase with H_2_O_2_ as an oxidizing agent [56], on the leaves of *S. bicolor* in response to salt stress. Consistent with MeJa’s role in reversing the deleterious effect of salt stress on the photosynthetic pigments of sorghum plants, MeJa-pre-treated *S. bicolor* plants showed substantial decreases in ROS accumulation under salt stress. These results illustrated MeJa’s role in mediating H_2_O_2_ scavenging under harsh conditions. The reduced ROS formation might be due to MeJa’s role in maintaining the stability of the chloroplast by protecting the photosystem II [57,58]. The ameliorating effect of MeJa on the over-accumulation of H_2_O_2_ in response to abiotic stress was also observed in *Glycyrrhiza uralensis* [55], *Vitis vinifera* L. [57], and *Zea mays* [59]. 

The cell membrane is a very important structure and houses intricate properties crucial for survival. Disruption in membrane properties can be deleterious due to the inhibition of ionic transport and eventually lead to cell death [60]. Increased lipid peroxidation, as indicated by high MDA content, was observed when *S. bicolor* plants were subjected to salt stress (Figure 2). Elevated levels of lipid peroxidation could be a result of limited stomatal conductance and increasing lipid membrane damage caused by ROS molecules due to osmotic stress [61]. These findings were previously supported by observations in *Oryza sativa* [62], *Ocimum basilicum* [63], *Zea mays* [59] and *Arachis hypogaea* [64] when exposed to salt stress. *S. bicolor* plants had significantly reduced levels of lipid peroxidation when pre-treated with 10 μM MeJa under salt stress, illustrating the dose-dependent ameliorating effect of MeJa on lipid peroxidation under harsh conditions. This ameliorating effect under stress conditions might be due to JA’s role in scavenging free radicals [65]. Decreasing MDA content under abiotic stress was also observed in *Brassica napus*, *Pisum sativum* L., and *Nitraria tangutorum* [38,65,66], in response to MeJa treatment. Pre-treatment with MeJa, therefore, played a beneficial role in maintaining cell membrane complexity under stressful conditions.

Assaying the amount of free proline found in plants is another method of assessing a plant’s tolerance to stress. Over-accumulation is often associated with severely perturbed metabolism whereby these molecules would aid protection against disease and stress mitigation in plants [60]. Salt stress disturbed osmotic potential and thereby triggered the over-accumulation of proline in *S. bicolor* plants (Figure 3A). Significant increments in proline content were observed under salt stress, illustrating a directly proportional relation between salt stress and proline content in *S. bicolor*. This increase is due to osmotic adjustment needed for plant survival under harsh environments [67]. The increasing levels of proline in response to adverse conditions are well documented, as observed in *Oryza satica*, *Zea mays,* and *Origanum vulgare* L. [67,68,69]. *S. bicolor* plants pre-treated with 10 μM MeJa significantly decreased proline content under salt stress conditions. However, the proline content assayed from plants treated with 15 μM MeJa remained at the same level as that of control (those treated with NaCl only) plants. Increasing proline is also generally accepted as it is correlated with decreasing the osmotic potential. This result, therefore, illustrated the role of MeJa in alleviating disturbance in osmotic potential and in maintaining cell turgor in *S. bicolor* plants under salt stress [50]. The decreasing levels of proline content under harsh environments were also observed in *S. bicolor* [16,36,37], *Anschusa italica* [51], *Zea mays* [59], *Pisum sativum* [65], and *Origanum vulgare* L [69].

In addition to proline, soluble sugars also play a crucial role in osmoregulation. Soluble sugars are crucial constituents of cellular respiration, making them an important energy source for plants [40]. Salt stress induced TSS content in *S. bicolor* plants (Figure 3B). The increasing soluble sugar content is important for osmotic adjustment and the scavenging of ROS molecules, and also aids in providing energy for survival under adverse conditions, as previously observed in *Triticum aestivum* L. [70], *Pisum sativum* [65], and *Vicia faba* L. [71]. Pre-treatment with 10 μM MeJa had no significant effect on TSS content under salt stress. However, 15 μM MeJa proved efficacious in reducing TSS content under 200 mM NaCl stress. The increasing effect of TSS in response to salt stress and the diminishing effect in MeJa pre-treated *S. bicolor* plants could be the result of a shift to the usage of secondary metabolites, or MeJa’s role in eliciting an early response to adverse conditions during pre-treatment [58,72]. It is well documented that exogenous application of MeJa increases soluble sugars under abiotic stress, but in this study the decreasing levels of TSS content due to 15 μM MeJa under salt stress conditions were consistent with those observed in *Glycyrrhiza uralensis* [55] and *Anchusa italica* [51].

Plant survival is highly dependent on systemic acquired adaptation to environmental stressors, and alterations of a plant’s epidermis is a crucial part of this systemic acclimation to environmental challenges [16]. The epidermis forms part of the first line of a plant’s defenses, preventing rapid water loss and excessive uptake of harmful substances [73,74,75]. It is therefore crucial in understanding the plant’s early responses and the regulation of stress-related alterations and adaptations [74,76,77]. Salt stress caused alterations to the epidermal and xylem tissue of *S. bicolor* plants (Figure 4B). This was indicated by increased shriveling on the epidermis and thinning of the xylem walls of *S. bicolor* plants under salt (200 mM NaCl) stress. The deformation and shrinkage observed in the epidermis of *S. bicolor* plants due to salt stress correlate with reduced growth, since previous reports showed that abiotic stress induces stiffening of plant epidermal cells, which may affect growth due to limited cell expansion [74]. Similarly, the vascular bundle (xylem and phloem) plays a major role in the transport of water and nutrients [44], thus any obstruction in their structure will affect plant growth. Thus, the observed stiffening and shrinkage of the xylem was another tolerance strategy employed by *S. bicolor* plants to limit excessive water loss [74,76,77,78]. MeJa had an ameliorating effect on the epidermis and xylem tissues of *S. bicolor* under salt stress by successfully preventing the stress-induced shrinkage and thinning of the anatomic structure.

Salt stress reduced the absorption and distribution of essential elements, particularly looking at K^+^ and Si^+^ under salt treatment as evidenced by high Na^+^/K^+^ (1.4) (Figure 4G,H; Table 3). MeJa improved element absorption and distribution, as seen by the reduced Na^+^ content, and this is supported by a very low Na^+^/K^+^ (0.16) ratio, especially under 15 μM MeJa pre-treatment. Furthermore, *S. bicolor* plants grown from MeJa-primed seeds showed increased Si^+^ absorption by 100% under salt stress compared to MeJa-untreated plants. These results suggest that exogenous MeJa promoted Si^+^ absorption in *S. bicolor* under salt stress by enhancing the plant’s ability to accumulate silica through activating silicon transporter genes [79,80]. Pre-treatment with MeJa, therefore, proved beneficial to the plant’s structure under adverse environments and enhanced support to the plant’s structure protecting the epidermis, and therefore allowing for optimal photosynthetic activity [79]. This was supported by the increased photosynthetic pigment in MeJa-pre-treated *S. bicolor* plants under salt stress.

The biochemical changes in the plant’s metabolism are highly dependent on the uptake and accumulation of inorganic and organic elements, which induces changes to lipids, proteins, and carbohydrates [80,81]. Salt stress caused major alterations in the functional groups of most molecules in *S. bicolor* plants (Figure 5), and changes in biomolecules might be related to initiating strategies for plant survival. These included phenolic compounds (3295 cm^−1^), such as flavonoids, phenolic acids or tannins [82]; alkyne-containing aliphatic compounds (2135 cm^−1^), such as acetylenic fatty acids [78]; aldehyde-containing aliphatic compounds (1733 cm^−1^), which may have been formed during lipid peroxidation [83]; and alkyl-halide-containing fluoro-compounds (1250 cm^−1^). Whereas the changes in primary or secondary amine-containing proteins (824 cm^−1^), correlates with the induced proline levels [84]; and the presence of halo compounds (551 cm^−1^), further suggest the induction in these compounds is related to plant stress tolerance.

*S. bicolor* plants pre-treated with MeJa had a substantial and clear alteration in their metabolic makeup under salt stress. An increase in the volume of phenolic compounds in *S. bicolor* plants grown from 10 μM MeJa-primed seeds under salt stress (200 mM NaCl) was observed, suggesting that induction in these secondary metabolites due to 10 μM MeJa is related to stress tolerance in *S. bicolor* plants. Additionally, FTIR spectroscopy identified decreased volumes of acetylenic fatty acids and primary or secondary amines as indicated by the decreased intensity in peaks found in frequency ranges of 2140–2100 cm^−1^ and 910–665 cm^−1^, respectively, in the metabolism of 10 µM MeJa pre-treated *S. bicolor* plants under salt stress. Slight changes in the metabolic network were observed in *S. bicolor* plants pre-treated with 15 μM MeJa in response to salt stress. This was indicated by the decreased volume of fatty acids (2135 cm^−1^) and primary or secondary amines (824 cm^−1^) identified in pre-treated *S. bicolor* subjected to harsh salt stress conditions.

The multidisciplinary approaches used in this study to investigate the role of MeJa on the stress tolerance of *S. bicolor* included comprehensive gene expression analysis to further elucidate stress mechanisms at a genetic level. Research has demonstrated that JA plays a crucial role in mediating the effects of abiotic stress and plant survival under unfavorable conditions [85]. To obtain a better understanding of the role of JA in alleviating salt stress effects in *S. bicolor*, this study focused on the changes in the transcription profile of JA-biosynthesis genes *SbLOX*, *SbAOS*, *SbAOC,* and *SbOPR* in *S. bicolor* plants under salt (200 mM NaCl) stress (Figure 6). All targeted genes *SbLOX*, *SbAOS*, *SbAOC,* and *SbOPR* were constitutively expressed, followed by significant upregulation in response to salt stress. The upregulation in the JA biosynthesis genes is related to the induction of tolerance in *S. bicolor*, as previously reported in transgenic *Arabidopsis thaliana* [86] and *Arachis hypogaea* [87] plants overexpressing JA genes. Pre-treatment with MeJa prevented the salt-induced upregulation in the expression of the JA-biosynthesis pathway genes in *S. bicolor* plants under adverse salt conditions. This was indicated by a significant downregulation in the transcript levels of *SbLOX* and *SbAOC* in *S. bicolor* plants grown from seeds primed with both MeJa (10 μM and 15 μM) concentrations, whereas only 15 μM MeJa led to decreased *SbAOS* and *SbOPR* transcripts in *S. bicolor* plants exposed to salt stress. However, 10 μM MeJa, which showed to be the most effective concentration in most assays, did not alter *SbAOS* transcript, whereas *SbOPR*’s transcript was further increased under salt stress. The reduced *SbLOX* and *SbAOC* transcripts under all MeJa treatments, and *SbAOS* and *SbOPR* under 15 μM MeJa, may be the result of a negative feedback loop arising through the induction of JASMONATE ZIM DOMAIN (JAZ) proteins inhibiting the activity of MYC2, which is a key activator for JA response as observed in transgenic *A. thaliana* [88,89]. The induced expression of JA-biosynthesis genes was also previously observed in response to salt stress [90,91,92]. However, the reduced expression of JA biosynthesis genes under high MeJa concentration (15 μM MeJa) is further supported by exogenous MeJa inhibiting JA biosynthesis through the inhibition of OPR activity in *Citrus sinensis* [91]. Contrary to this, this study showed a significant further increase in the transcript level of JA precursor gene *SbOPR* in *S. bicolor* plants grown from 10 μM MeJa-primed seeds under salt stress, which may indicate induced biosynthesis of JA. Furthermore, this supports the possibility that JA-independent pathways could regulate the synthesis of OPR precursor substrate 12-oxo phytodienoic acid (OPDA) [92,93].

## 4. Materials and Methods

### 4.1. Sorghum Germination and Growth Condition

*Sorghum bicolor* L. Moench (AgFlash/NIAGARA III: (Sorghum × Sudan), red seeds) were purchased from Agricol Brackenfell, Cape Town, South Africa. The *S. bicolor* seeds were surface decontaminated as previously described by Mulaudzi-Masuku et al. [94]. Briefly, seeds were washed with 70% ethanol followed by soaking in 5% sodium hypochlorite solution (NaOCl) for 1 hour while shaking at 600 rpm. This was followed by washing thrice with autoclaved ddH_2_O. After surface decontamination, seeds were left to imbibe by soaking in autoclaved ddH_2_O overnight at 25 °C while shaking in the dark. After imbibition, ddH_2_O was removed and the decontaminated seeds were placed on sterile paper to dry at room temperature under the laminar flow until their original moisture content was obtained. Germination was induced by incubating the dried seeds on a sterile water-imbibed sterile paper towel for 7 days at 25 °C while being completely concealed from light. After 7 days of incubation, seedlings were transferred into seedling trays containing 35 g potting soil and vermiculite (2:1). Each seedling was irrigated with 25 mL Dr Fisher’s Multifeed Classic 19:8:16 (43) (Gouws and Scheepers Pty, Ltd., London, UK) every alternate day until the plantlets reached their 3-leaf stage of development. Seedlings were grown under summer conditions where temperatures ranged from 22 °C to 26 °C. Plantlets were harvested when the 5-leaf stage of development was reached. During harvesting the shoots were carefully separated and stored at −80 °C until required for assays. Immediately after separation, plant materials were weighed to determine fresh weight followed by drying in an oven at 80 °C for 24 h to determine dry weight.

### 4.2. Treatment Application

After reaching the 3-leaf stage of growth in the soil, plantlets were stressed with different salt concentrations, including 0 mM NaCl (serving as the control) and 200 mM NaCl (serving as the salt treatment). The plants were stressed until reaching the 5-leaf stage of development after 7 days. During the imbibition step (as described in Section 2.1), seeds were pre-treated by priming them with different concentrations of MeJa in 50 mL solutions containing 0 μM (control), 10 μM, and 15 μM MeJa concentrations overnight at 25 °C while shaking in total darkness.

### 4.3. Histochemical Detection of H_2_O_2_

Over-accumulation of hydrogen peroxide (H_2_O_2_) was determined by histochemical staining based on its role in oxidizing 3,3′-diaminobenzidene (DAB) in a reaction induced by horseradish peroxidase to form brown formazan at the location of enzyme activity [56]. Hydrogen peroxide localization was determined in the leaves as previously described [12,95]. Briefly, leaves were submerged in 1 mg/mL 3′,3′ diaminobenzidine (DAB), pH 3.8, for 12 h at 25 °C while concealed from light. Following incubation, chlorophyll was extracted by boiling the leaves in 90% ethanol for 15 min for clear identification of formazan caused by the oxidation of DAB by H_2_O_2_.

### 4.4. Lipid Peroxidation

Lipid peroxidation is indicative of the oxidation of unsaturated fatty acids in the cell membrane, whereby malondialdehyde (MDA) is produced as a by-product [12,96]. Malondialdehyde accumulation was selected as a biomarker of lipid peroxidation in this study, following the traditional thiobarbituric acid (TBA) reaction method to measure MDA content as previously determined with slight modifications [97]. Macerated plant material (0.1 g) was homogenized with 0.1% TCA (1 mL) and centrifuged for 10 min at 13,000 rpm (4 °C). About 1 mL of 0.5% TBA prepared in 20% TCA was added to 0.4 mL of supernatant after centrifugation. To avoid pressure build-up, caps of the Eppendorf tubes were perforated before being incubated in a hot water bath set at 80 °C for 30 min. The reaction was terminated by immediately placing the samples on ice after incubation, followed by further centrifugation for 5 min at 13,500 rpm (4 °C). Absorbance readings were measured at 532 nm and corrected for nonspecific turbidity by subtracting absorbance from 600 nm using the Helios^®^ Epsilon visible 8 nm bandwidth spectrophotometer (Thermo Fisher Scientific, Waltham, MA, USA).

### 4.5. Chlorophyll Content

Chlorophyll and carotenoid content were determined based on previously described methods [98,99]. Briefly, 0.1 g of ground plant material was homogenized in 10 mL of 80% acetone. The homogenates were thoroughly vortexed and centrifuged at 10,000 rpm for 10 min. About 1 mL was aliquoted into a cuvette and the pigment absorption was spectrophotometrically measured at 645 nm and 663 nm (Mackinney, 1968) using the Helios^®^ Epsilon visible 8 nm bandwidth spectrophotometer (Thermo Fisher Scientific, USA). Chlorophyll content was calculated according to Arnon (1949) as follows:Chlorophyll a = 12.7 (A663) − 2.69 (A645)
Chlorophyll b = 22.9 (A645) − 4.68 (A663)
Total Chlorophyll = 20.2 (A645) + 8.02 (A663)

### 4.6. Proline Content

Proline levels were measured as previously described with slight modifications [100]. Macerated plant material (0.1 g) was homogenized with 3% sulfosalicylic acid (0.5 mL), vigorously mixed, and centrifuged for 20 min at 13 000 rpm. The supernatant (0.3 mL) was transferred to a reaction mixture containing 99% glacial acetic acid and acidic ninhydrin solution (2.5% ninhydrin, 60% acetic acid and 20% ethanol). The reaction was induced by heating the samples in a hot water bath set at 95 °C for 20 min and arrested by placing them on ice, followed by further centrifugation at 13,000 rpm. The chromophore (200 μL) was transferred to a 96-well microplate and its absorbance was measured at 520 nm using the FLUOstar^®^ Omega microtiter plate reader (BMG LABTECH, Ortenberg, Germany). Proline concentration was determined according to a calibration curve containing known proline concentrations and expressed as μmol g^−1^ FW.

### 4.7. Total Soluble Sugars

Total soluble sugar content was determined as previously described in the anthrone method with some modifications [101]. Ground plant material (0.1 g) was homogenized with 80% acetone and centrifuged for 10 min at 10,000 rpm. About 1 mL of the supernatant was transferred to a 0.2% anthrone solution (0.2 g anthrone dissolved in 96% sulfuric acid) and placed in a hot water bath for 15 min at 80 °C. The reaction was terminated by placing the samples on ice. Thereafter, 1 mL of the sample was aliquoted to a glass cuvette and its absorbance was measured at 625 nm. Levels of total soluble sugars were quantified using a calibration curve containing known glucose concentrations and expressed as mg·g^−1^ FW.

### 4.8. Anatomic Analysis Using Scanning Electron Microscopy

The anatomic structure, including the epidermis, xylem, and element distribution, was analyzed at the University of the Western Cape using high-resolution scanning electron microscopy (HRSEM) as described previously [12,37,64]. Microphotographs were taken to obtain the morphological characteristics using the Tescan MIRA field emission gun scanning electron microscope using an in-lens secondary electron detector and set to an operating acceleration voltage of 5 kV. All spectra were analyzed using the built-in Oxford Inca software suite.

### 4.9. Fourier-Transform Infrared (FTIR) Spectroscopic Analysis of Biomolecules

Samples were analyzed using the Perkin Elmer Spectrum 100-FTIR Spectrometer [PerkinElmer (Pty) Ltd., Midrand, South Africa] as previously described [37]. Briefly, 2 g of dried macerated plant material was homogenized with 0.4 g of a pre-dried KBr, followed by scanning of the pellet mixture (~2 g). A wider spectral window frequency range of 450 to 4000 cm^−1^ was considered.

### 4.10. RNA Extraction and cDNA Preparation

Total RNA was extracted from 0.1 g of macerated plant material (treated and untreated) using the FavorPrep TM Plant Total RNA Purification Mini Kit (FAPRK001-1, Favorgen Biotech Corp, Ping-Ting, Taiwan) following the manufacturer’s protocol. The genomic DNA was removed by treating the resulting RNA with an RNase-free DNase solution (New England Biolabs, Massachusetts, MA, USA). Complementary DNA was synthesized from ~1 μg of extracted RNA using the SuperScript TM III First-Strand Synthesis kit (Invitrogen, Carlsbad, CA, USA) following the manufacturer’s protocol. The high quality and integrity of both RNA and cDNA were spectrophotometrically assured by quantification using NanoDropTM 2000/2000c spectrophotometer (Thermo Scientific, Waltham, MA, USA).

### 4.11. Quantitative Real-Time Polymerase Chain Reaction

Quantitative Real-Time Polymerase Chain Reaction (qRT-PCR) was used to determine expression profiles using the LightCycler^®^ 480 SYBR Green I Master kit (Roche Diagnostics, SA) following the manufacturer’s protocol. One reaction contained 1 μL DNA template, 5 μL 2× SYBR Green I Master Mix (Roche Applied Science, Germany) and optimized primer concentrations made up to a final volume of 10 μL with RNase-free dH_2_O. Primers were designed using the Primer3 online tool (https://primer3.ut.ee; accessed 3 May 2019) and the primer sequence is shown in Table 4. The reactions were subjected to 95 °C for 10 min, 47 cycles at 95 °C for 10 s, 58 °C for 10 s (*SbLOX; SbAOS; SbAOC; SbOPR*), 60 °C for 10 s (PEPC; Beta actin) at 40 cycles, 60 °C for 10 s (UBQ) at 46 cycles, and 72 °C for 20 s. A melting curve was performed using default parameters. Primer information is listed below. Transcript expression levels were normalized using Beta actin, Ubiquitin (UBQ) and Phosphoenolpyruvate carboxylase (PEPC) as reference genes [94] and analyzed using the LightCycler^®^ 480 Software version 1.5.1.62. Expression levels were quantified relative to a calibration curve of serially diluted cDNA. The results obtained serve as a representation of 3 biological and 3 independent repeats including non-template control.

### 4.12. Statistical Analysis

All experiments were repeated at least three times and data were statistically analyzed by a one-way ANOVA using Minitab^®^ 21 Statistical Software (https://www.minitab.com/en-us/support/downloads/ accessed on 24 November 2022). Data in the Figures and Tables represent the mean ± standard deviation from three biological replicates. Statistical significance between control and treated plants was determined by Tukey’s test for comparison at a 95% confidence interval. The difference was regarded as significant when *p* < 0.05. Means that do not share a letter in a column were significantly different.

## 5. Conclusions

In this study, we demonstrated that priming *S. bicolor* seeds with MeJa enhances salt stress tolerance. Based on the data obtained, we showed that morphologically, *S. bicolor* is sensitive to salt stress at the seedling stage. This was demonstrated by the reduced growth and biomass. However, priming seeds with MeJa proved that induction of endogenous JA in cells provides some sort of protection and that the concentration is dependent on the severity of the stress, e.g., 15 μM MeJa was more effective compared to 10 μM MeJa for the 200 mM NaCl-stressed *S. bicolor* plants in maintaining the growth of *S. bicolor* plants under stress to the same level as that of the control (without NaCl treatment). Furthermore, both MeJa concentrations led to improved photosynthetic performance, reduced oxidative stress, and improved osmoregulation under both salt concentrations. Furthermore, MeJa prevented salt-induced physiological and biochemical changes by reducing the shrinkage and thinning of the epidermis and xylem tissues. This correlated with improved element absorption as confirmed by the reduced Na^+^/K^+^ ratio and 100% restoration of Si^+^ absorption, proving that MeJa played a role in either the exclusion or compartmentalization of toxic Na^+^ ions. Additionally, MeJa altered the structure of biomolecules by further inducing the synthesis of secondary metabolites. Lastly, MeJa alleviated salt stress and was also assayed based on the expression levels of JA precursor gene *SbOPR*. We, therefore, propose a preliminary mechanism, and suggest that exogenous MeJa mediated salt tolerance in *S. bicolor* by inducing the synthesis of endogenous JA through increased *OPR* transcript under salt stress, therefore activating other signaling cascades including ROS scavenging capacity, osmoregulation, and ion transport to maintain a steady homeostasis and hence improve growth under salt stress. Data from this study will pioneer the establishment of transgenic crop cultivars with improved tolerance and therefore increased crop productivity. Further analyses are required on the effect of exogenous MeJa on the downstream molecules in the JA pathway and investigation of a possible JA-independent pathway playing a role in inducing endogenous JA.

## Figures and Tables

**Figure 1 ijms-24-10368-f001:**
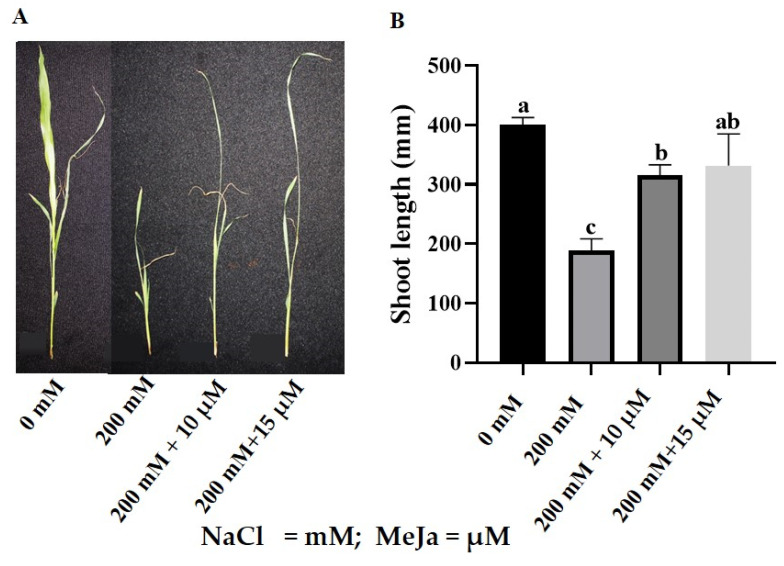
Effect of MeJa on the growth of *S. bicolor* under salt stress. Phenotype (**A**) and shoot length (**B**) of *S. bicolor* plants grown from seeds primed with 10 µM and 15 μM MeJa and exposed to 200 mM NaCl. Data in the figure represent the mean ± standard deviation from three biological replicates. Different letters indicate significant differences (*p* < 0.05) based on ANOVA one-way variance analysis following Tukey’s comparison test.

**Figure 2 ijms-24-10368-f002:**
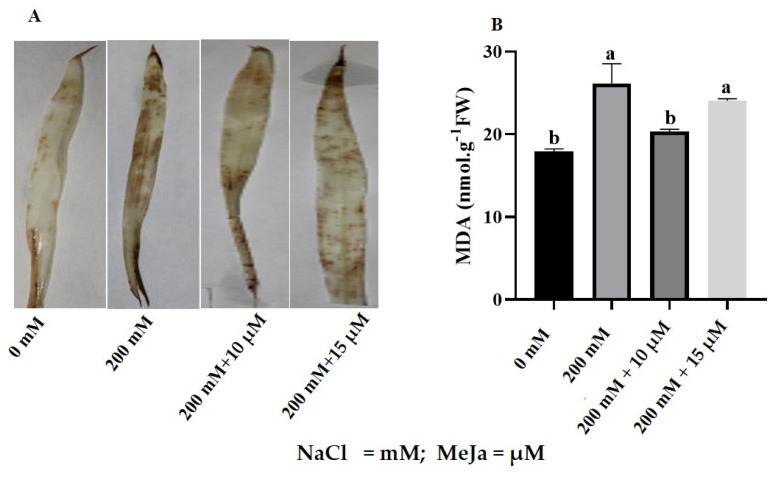
Assessment of oxidative damage in *S. bicolor* in response to MeJa under salt stress. Accumulation of H_2_O_2_ on the leaves was determined by histochemical staining (**A**), whereas lipid peroxidation was determined by measuring MDA content (**B**) under 200 mM NaCl for *S. bicolor* plants grown from seeds primed with 10 µM and 15 µM MeJa. Data in the figure represent the mean ± standard deviation from three biological replicates. Different letters indicate significant differences (*p* < 0.05) based on ANOVA one-way variance analysis using Tukey’s comparison test.

**Figure 3 ijms-24-10368-f003:**
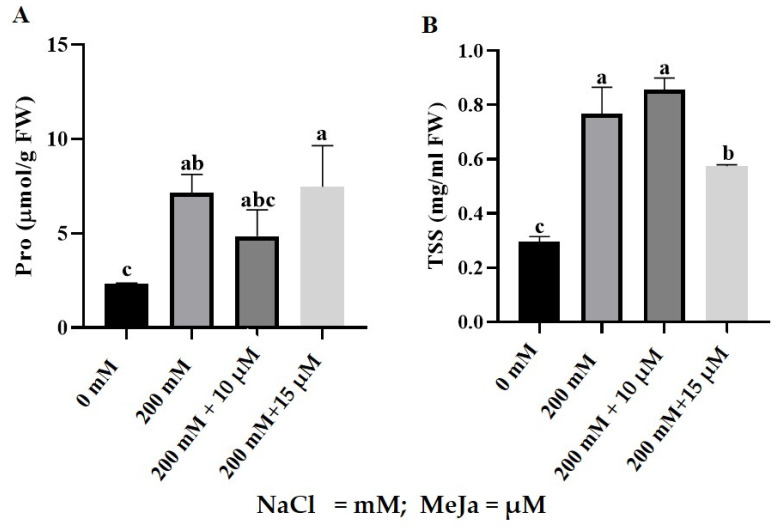
Proline (Pro) and total soluble sugar (TSS) content in response to MeJa pre-treatment in *S. bicolor* under salt stress. Proline content (**A**) and total soluble sugars (**B**) in *S. bicolor* plants grown from seeds primed with 10 and 15 µM MeJa and exposed to 200 mM NaCl. Data in the figure represent the mean ± standard deviation from three biological replicates. Different letters indicate significant differences (*p* < 0.05) based on ANOVA one-way variance analysis using Tukey’s comparison test.

**Figure 4 ijms-24-10368-f004:**
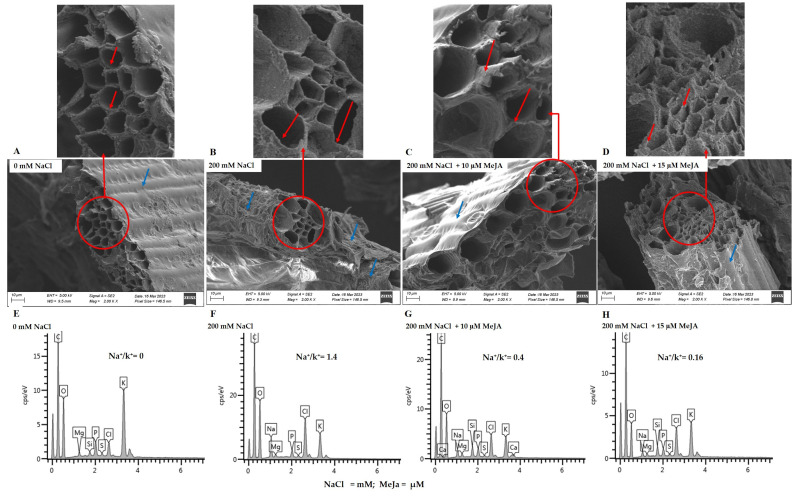
Scanning electron microscopy images illustrating the epidermis, xylem, and element distribution in *S. bicolor* plants grown from seeds primed with MeJa and exposed to salt stress. Cross-section images of *S. bicolor* epidermis and xylem (**A**–**D**), and EDX spectrum (**E**–**H**), obtained from plants treated with 0 mM NaCl (**A**,**E**), 200 mM NaCl (**B**,**F**), 200 mM NaCl + 10 µM MeJa (**C**,**G**), and 200 mM NaCl + 15 µM MeJa (**D**,**H**). Epidermis and xylem are highlighted by blue and red arrows, respectively. For clarity purposes a small area showing the xylem walls in the SEM micrographs has been selected and enlarged/zoomed.

**Figure 5 ijms-24-10368-f005:**
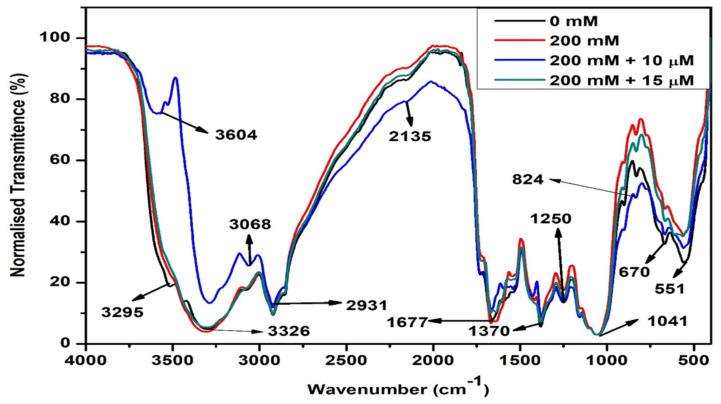
FTIR analysis of the effect of pre-treatment with MeJa on the biomolecules in *S. bicolor* under salt stress conditions. Changes in functional groups were detected in control *S. bicolor* plants (black) and pre-treated with 0 (red), 10 (blue), and 15 µM (green) of MeJa under 200 mM NaCl stress conditions [NaCl = mM; MeJa = µM].

**Figure 6 ijms-24-10368-f006:**
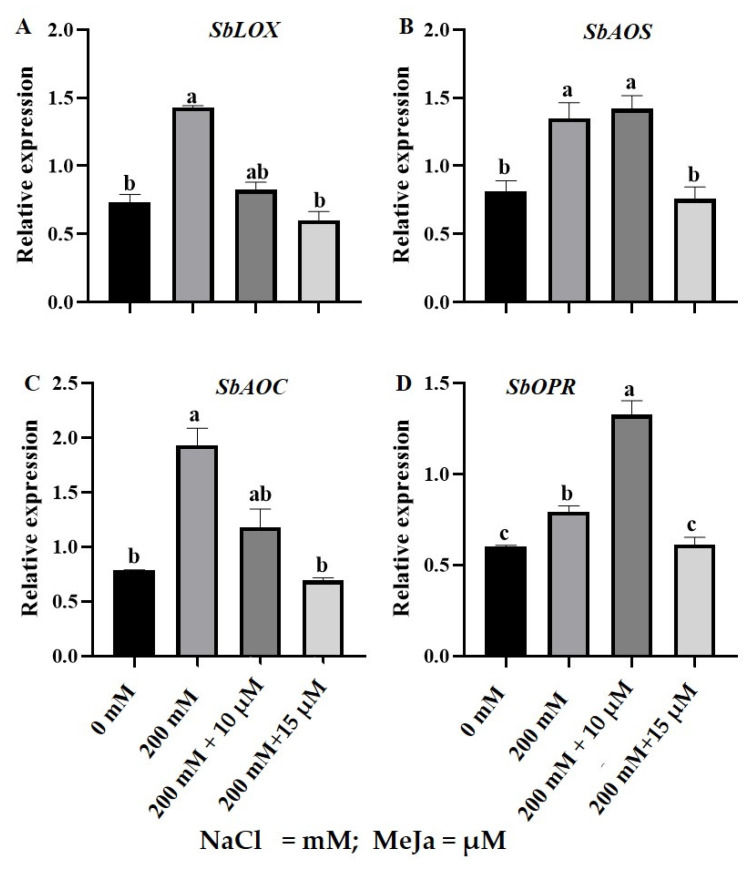
The relative expression levels of JA biosynthesis genes in *S. bicolor* pre-treated with 10 µM or 15 µM MeJa under salt stress (200 mM NaCl). Relative expression levels of *SbLOX* (**A**), *SbAOS* (**B**), *SbAOC* (**C**), and *SbOPR* (**D**) were analyzed from the total RNA extracted from *S. bicolor* plants using qRT-PCR. Data in the figure represent the mean ± standard deviation from three biological replicates. Different letters indicate significant differences (*p* < 0.05) based on ANOVA one-way variance analyzed using Tukey’s comparison test.

**Table 1 ijms-24-10368-t001:** Plant biomass reflecting the effect of MeJa in *S. bicolor* under salt stress. Fresh weight and dry weight were determined from *S. bicolor* plants grown from seeds primed with 10 µM, and 15 µM MeJa and stressed with 200 mM NaCl. Data in the table represent the mean ± standard deviation from three biological replicates. Different letters indicate significant differences (*p* < 0.05) based on ANOVA one-way variance analysis followed by Tukey’s comparison test.

NaCl (mM)	MeJa (µM)	Fresh Weight (g)	Dry Weight (g)
0	0	0.800 ± 0.189 ^a^	0.091 ± 0.013 ^a^
200	0	0.397 ± 0.027 ^b^	0.053 ± 0.005 ^b^
	10	0.730 ± 0.031 ^a^	0.110 ± 0.005 ^a^
	15	0.665 ± 0.039 ^ab^	0.102 ± 0.00 ^ab^

**Table 2 ijms-24-10368-t002:** Effect of Methyl Jasmonate (MeJa) on the chlorophyll content in *S. bicolor* under salt stress. Photosynthetic pigments (Chlorophyll *a*, Chlorophyll *b* and total Chlorophyll) were determined from *S. bicolor* plants grown from seeds primed with 10 and 15 µM MeJa and stressed with 200 mM NaCl. Data in the table represent the mean ± standard deviation from three biological replicates. Different letters in the column indicate significant differences (*p* < 0.05) based on ANOVA one-way variance analysis using Tukey’s comparison test.

NaCl (mM)	MeJa (µM)	Chlorophyll *a* (mg/gFW)	Chlorophyll *b* (mg/gFW)	Total Chlorophyll (mg/gFW)
0	0	8.17 ± 1.890 ^ab^	4.392 ± 1.075 ^a^	11.96 ± 2.98 ^ab^
200	0	4.308 ± 0.949 ^b^	3.172 ± 0.820 ^b^	6.667 ± 2.078 ^b^
	10	11.992 ± 3.440 ^a^	5.011 ± 0.516 ^a^	16.710 ± 3.900 ^a^
	15	9.104 ± 0.656 ^ab^	4.585 ± 0.513 ^ab^	13.438 ± 0.151 ^a^

**Table 3 ijms-24-10368-t003:** Element distribution in *S. bicolor* plants in response to MeJa priming under salt stress. Elements were analyzed from *S. bicolor* plants that were grown from seeds primed with 10 and 15 µM MeJa and exposed to 200 mM NaCl. Data in the table represent the mean ± standard deviation from three biological replicates. Different letters in the column indicate significant differences (*p* < 0.05) based on ANOVA one-way variance analysis using Tukey’s comparison test.

Elements	0 mM NaCl (wt%)	200 mM NaCl (wt%)	200 mM + 10 µM MeJa (wt%)	200 mM + 15 µM MeJa (wt%)
Na^+^	0.00 ± 0.00 ^c^	2.16 ± 0.52 ^a^	0.97 ± 0.24 ^b^	0.635 ± 0.155 ^bc^
Si^+^	0.295 ± 0.095 ^a^	0.00 ± 0.00 ^b^	0.72 0.24 ^ab^	0.675 ± 0.225 ^ab^
K^+^	7.46 ± 3.38 ^a^	1.49 ± 0.17 ^b^	2.53 ± 1.2 ^ab^	4.17 ± 1.97 ^ab^
Element Ratio				
Na^+^/K^+^	0.00 ± 0.00 ^c^	1.436 ± 0.186 ^a^	0.419 ± 0.117 ^b^	0.166 ± 0.047 ^bc^

**Table 4 ijms-24-10368-t004:** Information of the JA biosynthesis pathway genes studied under salt stress in response to priming with MeJa.

Gene Name	Accession Numbers		Amplicon Size		Primer Sequence
	LOC ID	mRNA Sequence ID		Forward (5′-3′)	Reverse (5′-3′)
*SbLOX*	LOC8064218	GQ369443.1	204 bp	GTACCGCTACGACGTCTACA	GTCAACTCTCGTGCAGCAAA
*SbAOS*	LOC8086184	XM_002463784.2	191 bp	ACCATCACCTCGCTCAAGAA	TCACACAGTATCACGGCACT
*SbAOC*	LOC8063218	XM_002465042.2	161 bp	GTACGAGGCCATCTACAGCT	AGGGGAAGACGATCTGGTTG
*SbOPR*	LOC8070775	XM_002438007.2	166 bp	GGGTATGATCGGGAGGAAGG	CAACGGGATCTTGCGTGTAG

## Data Availability

No new data are available.

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
