# Peer review of "Seed Priming with MeJa Prevents Salt-Induced Growth Inhibition and Oxidative Damage in Sorghum bicolor by Inducing the Expression of Jasmonic Acid Biosynthesis Genes"

_ijms, 2023, doi:10.3390/ijms241210368_

Round 1

Reviewer 1 Report

Dear authors, thank you for the work dealing with approaches for the yeild improvement in Sorghum.

The presented manuscript demonstrates data on the role of  seeds priming with MeJA in acclimation plants to salt-stress. Much attention is paid to the physiological and biochemical changes in plants, grown from primed and non-primed seeds under the action of NaCl. It has been shown that priming seeds with MeJa in some cases has positive effects on plants, which is associated with a decrease in the level of LPO and the content of hydrogen peroxide, and an increase in the content of proline and free sugars.

The planning of the experiment is not perfect. There is no control with methyl jasmonate (treated seeds, but growth without NaCl), so it is not entirely correct to draw conclusions about the role of MeJa in salt stress. When assessing the role of MeJa, it is better to discuss data based on dry weight, and not on fresh weight, since the water content and water status under salt stress vary greatly.

The part of the work describing the anatomical structure of the leaves in control and under stress or MeJa treatment is difficult to understand. The quality of the images does not allow to verify the structures noted by the authors. The figures do not indicate the scale and magnification.

The section including data on expression also raises questions. It is not discussed in any way what caused the change in the expression level of methyl jasmonate synthesis genes in leaves many days after seed treatment.

It is not clear why different indicators are defined for plants in different versions of the experiment. Somewhere there is only 1 concentration of NaCl, somewhere two. Somewhere some concentrations of MeJa, somewhere others.

It is not specified which variety or line of sorghum was used in the experiment.

Table 3 does not specify the units of measurement of the indicator.

In the statistics section, it is not clear what n=3 means. Is it 1 plant in three experiments (the authors write that they were repeated three times) or 3 plants in one experiment? Then how are the repetitions of experience considered in the statistics?

The results section provides a lot of information that should have been in the Introduction or in the Methods. Discussion is very week. The Discussion should reveal the links between the effects found and explain them using literature data. The authors just write that other researchers revealed similar effects on other plants.

Thus, the submitted manuscript cannot be published in current state.

English requires serious corrections.

Author Response

Dear editor

We are grateful for the reviewers who extensively reviewed our manuscript and provided critical reports, which in turn improved our manuscript. Below are point-by-point authors responses to the comments and as implemented in the revised manuscript.

Reviewer 1

English editing

Dear Reviewer kindly note we have extensively revised the English in the manuscript.

Reviewer comment #1: The planning of the experiment is not perfect. There is no control with methyl jasmonate (treated seeds, but growth without NaCl), so it is not entirely correct to draw conclusions about the role of MeJa in salt stress.

Response: The overall study included control plants (without either salt or MeJa treatments), plants (without salt treatments), but primed with different MeJa concentrations [(10 µM, 15 µM and 20 µM (20 µM is not included in this paper)]. Only assays on growth, biomass, chlorophyll, H2O2, MDA, proline, and total soluble sugars, were conducted on the control plants treated with MeJa (Results shared in supplementary data). Since there were no significant differences between control and 10 µM MeJa primed plants, whereas higher concentration (15 µM), showed slight negative effects, this data was therefore excluded in our original manuscript, and also since our main focus was to investigate the role of exogenous MeJA to protect sorghum seedlings from the effects of salt stress. Furthermore, these plants were not subjected to SEM-EDX, FTIR analysis and gene expression, since these assays are expensive.

Reviewer comment #2: When assessing the role of MeJa, it is better to discuss data based on dry weight, and not on fresh weight, since the water content and water status under salt stress vary greatly.

Response: We have taken note of this comment. But, since both fresh weight and dry weight showed good results [Table 1, there is significant difference under 200 mM NaCl treatments in the absence and presence of MeJa] we felt that it is important to include fresh weight results.

Reviewer comment #3: The part of work describing the anatomical structure of the leaves in control and under stress or MeJa treatment is difficult to understand. The quality of the images does not allow to verify the structures noted by the authors. The figures do not indicate the scale and magnification.

Response: Magnification of the anatomical structure has been added and the image quality has been updated.

Reviewer comment #3: It is not specified, which variety or line of sorghum was used in the experiment

Response: Variety “sweet sorghum (red seeds) was used for this study.

Reviewer comment #4: Table 3 does not specify the units of measurement of indicator.

Response: It was an oversight. We have now included the units (wt%).

Reviewer comment #5: In the statistics section, it is not clear what n=3 mean?

Response: Each data represents the mean taken from 3 independent biological replicates [same experiment conducted thrice under the same conditions to obtain reproducibility], each sample from the 3 replicates represent a few plants pooled to obtain enough material.

Reviewer comment #6: The section including the data on expression also raise questions. It is not discussed in any way what caused the change in the expression level of methyl jasmonate synthesis genes in leaves many days after seed treatment.

Response: The SbLOX, SbAOS, SbAOC and SbOPR genes expression levels were upregulated in the presence of salt stress since these genes are constitutively expressed in plants. However, priming with MeJa led to a decrease in the transcript levels of these genes which might be as a result of negative feedback through the induction of Jasmonate ZIM domain (JAZ) which inhibits the activation of JA responses by the activity of MYC2 as discussed. 

Reviewer comment #7: It is not clear why the different indicators are defined for plants in different versions of the experiment. Somewhere there is only 1 concentration of NaCl, somewhere two. Somewhere some concentrations of MeJa, somewhere others.

Response: The experiment was conducted with two salt concentrations (100 & 200 mM NaCl) and two concentrations of MeJa (10 & 15 µM). Some assays including shoot length, biomass, Chlorophyll pigments, ROS, MDA, proline and total soluble sugars samples harvested from both salt (100 mM and 200 mM NaCl) concentrations were analysed. Due to the effectiveness of which was narrowed to one salt concentration (200 mM NaCl) based on the effective role of MeJa on 200 mM NaCl-stressed plants, and the cost of the assays, for Anatomical and gene expression assays only 200 Mm NaCl was considered.

Reviewer comment #8: The results section provides a lot of information that should have been in the Introduction or in the Methods

Response: Results section has been updated as indicated by the reviewer.

Reviewer comment #9: Discussion is very weak. The Discussion should reveal the links between the effects found and explain them using literature data.

Response: Discussion has been revised. Reasons for observed effects of salt and MeJa have been explained.

Reviewer 2 Report

The paper titled"Seed priming with MeJA prevents salt-induced growth inhibition and oxidative damage in Sorghum bicolor by inducing the expression of Jasmonic acid biosynthetic genes", the authors showed that salt stress induced oxidative damage but priming with MeJA improved growth and prevent the oxidative damage under salt stress.

1. For the salt stress treatment, did the plants look happy and healthy? how about the phenotype after further recovery of the plants from stress?

2. MeJA as an exogenous hormone, did the author tried the treatment of MeJA only? Are there any growth phenotype?

3. Normally, as plants getting old, the resistance to stresses increase, did the authors tried different ages of the plant seedlings?

4. In Figure 1B, Salt1+MeJA1 looks have higher shoots compared to Salt1 only treatment, how many plants are used in the treatment and for the quantitative analysis?

5. Are there any of the genes expressed in both salt and MeJA pathway that helps coordinate the stresses?

Author Response

Dear editor

We are grateful for the reviewers who extensively reviewed our manuscript and provided critical reports, which in turn improved our manuscript. Below are point-by-point authors responses to the comments and as implemented in the revised manuscript.

Reviewer 2

Reviewer comment #1: For the salt stressed treatment, the plants looked healthy For the salt stress treatment, did the plants look happy and healthy? How about the phenotype after further recovery of the plants from stress?

Response: Plants under salt stress looked healthy, but the pictures taken are not of good quality. But as highlighted under section 2.1.1, sorghum plants showed reduction of shoot length under salt stress as shown in Figure 1A. The phenotype of the MeJa-primed plants showed that the growth inhibition by salt was prevented (Figure 1B & C), but their phenotype is not comparable to those under salt stress only.

Reviewer comment #2. MeJA as an exogenous hormone, did the author tried the treatment on MeJA only? Are there any growth Phenotype?

Response: Treatment with MeJa only was also conducted and the results are included as supplementary data. Based on the results, only 10 µM showed similar phenotype with the control (0 mM NaCl), whereas high concentration (15 µM) showed slight defects on the phenotype.

Reviewer comment #3. Normally, as plants getting old, the resistance to stresses increase, did the authors tried different ages of the plant seedlings?

Response: No, we only focused on the vegetative stage of sorghum since in the study we wanted to investigate response at early stage.

Reviewer comment #4: In Figure 1B, Salt1 + MeJA1 looks have higher shoots compared to salt1 only treatment.

Response: It is correct, plants under the Salt1 + MeJA1 (now changed to 100 mM NaCl + 10 µM) showed high shoot length as compared to those in Salt1 (now changed to 100 mM NaCl) only treatment. This is expected since the addition of MeJA has positive effects.

Reviewer comment #4: How many plants are used in the treatment and for the quantitative analysis?

Response: We grow as many plants as possible, and for the assays and statistical analysis we choose 3 plants, with similar phenotype, and this represent one biological replicate. The experiment is then repeated again twice following the same conditions, to obtain three independent biological replicates, thus in total 9 plants are selected for one experiment.

Round 2

Reviewer 1 Report

Dear authors, thank you for the work done on your manuscript.

It is much clear in metodology.

I suppose the name "sweet sorghum (red seeds)" is not enough for research paper. Sweet sorghum is any of the many varieties  with a high sugar content.  It is important to know the name of variety or the number of the hybrid. 

I believe that it will not be difficult to make the appropriate adjustments 

The quality of English is much better, but still needs corrections, mainly  inserted new fragments.

Author Response

Dear Reviewer

We are grateful for the reviewers for extensively reviewing our manuscript and provided critical reports, which in turn improved our manuscript. Below are point-by-point authors responses to the comments and as implemented in the revised manuscript.

Reviewer 1

Reviewer comment #1: Dear authors, thank you for the work done on your manuscript.

It is much clear in methodology.

I suppose the name "sweet sorghum (red seeds)" is not enough for research paper. Sweet sorghum is any of the many varieties with a high sugar content.  It is important to know the name of variety or the number of the hybrid. 

I believe that it will not be difficult to make the appropriate adjustments. 

Response: The cultivar name is AgFlash/NIAGARA III: (Sorghum x Sudan. This information has been updated in the main document.

Reviewer comment #2. Comments on the Quality of English Language

The quality of English is much better, but still needs corrections, mainly inserted new fragments.

Response: The manuscript has been revised and the language improved, paying more attention on the new sections.
